# Diabetes Related Distress in Children with Type 1 Diabetes before and during the COVID-19 Lockdown in Spring 2020

**DOI:** 10.3390/ijerph18168527

**Published:** 2021-08-12

**Authors:** Beata Mianowska, Anna Fedorczak, Arkadiusz Michalak, Weronika Pokora, Inga Barańska-Nowicka, Monika Wilczyńska, Agnieszka Szadkowska

**Affiliations:** 1Department of Pediatrics, Diabetology, Endocrinology and Nephrology, Medical University of Lodz, 90-419 Lodz, Poland; arkadiusz.michalak.lek@gmail.com (A.M.); agnieszka.szadkowska@umed.lodz.pl (A.S.); 2Department of Endocrinology and Metabolic Diseases, Polish Mother’s Memorial Hospital and Research Institute, 93-338 Lodz, Poland; anna.fedorczak@outlook.com; 3Department of Anaesthesiology and Intensive Therapy, Medical University of Lodz, 90-419 Lodz, Poland; 4Department of Biostatistics and Translational Medicine, Medical University of Lodz, 92-215 Lodz, Poland; 5Student Research Group of Department of Pediatrics, Diabetology, Endocrinology and Nephrology, Medical University of Lodz, 90-419 Lodz, Poland; pokora.weronika@gmail.com (W.P.); monika.wilczynska97@gmail.com (M.W.)

**Keywords:** COVID-19, diabetes distress, child, type 1 diabetes mellitus, pandemic

## Abstract

Our aim was to compere diabetes-related distress (DD) in young patients with type 1 diabetes mellitus (T1DM) and in their parents before and during the national COVID-19-related lockdown when schools operated on-line. Problems Areas in Diabetes-Child (PAID-Ch), Teen (PAID-T) and Parent (P-PAID-Ch, P-PAID-T) questionnaires in paper version were used to evaluate DD before COVID-19 pandemic (November 2019–February 2020) and during the lockdown (April 2020) the same surveys were performed by phone. We enrolled 76 patients (median age (Q1–Q3): 13.6 (11.8–15.2) years; 21 children, 55 adolescents; T1DM duration 3.7 (1.7–6.8) years). Initial PAID score was lower in teenage boys than in girls (34.0 (24.0–42.0) vs. 44.5 (40.0–50.5), *p* = 0.003). In teens PAID score decreased significantly during the lockdown (−3.0 (−11.0–3.0), *p* = 0.018), more in girls than boys (*p* = 0.028). In children (−3.0 (−14.0–7.0), *p* = 0.131) and parents PAID did not change (teens’ parents: 3.0 (−9.0–10.0), *p* = 0.376; children’s parents: −5.0 [−9.0–1.0], *p* = 0.227). In the studied group COVID-19 pandemic-related lockdown was associated with decrease in DD in teens with T1DM, particularly in girls, while no significant change in DD was observed in children or parents. DD decrease in teens during the pandemic should attract attention to the potential “rebound” of DD related to return to regular on-site school routine.

## 1. Introduction

Children with type 1 diabetes (T1DM) and their parents or caregivers are at elevated risk for psychological problems and one of the most important factors responsible for psychological burden in these groups is diabetes distress (DD) [1,2,3]. DD is related to diabetes regimen-specific duties with all therapeutic activities and decisions made by patients and their caregivers (i.e., frequent glucose monitoring with continuous glucose monitoring systems and/or with glucose meters, assessing carbohydrates or calories in meals, adjusting food and insulin doses to everyday situations, insulin injections/operating insulin pump), along with social distress (e.g., sense of loneliness amongst peers or family due to diabetes, deficiency of social backup or support), diabetes-related fears (e.g., fear of hypoglycaemia or potential future complications) and other patients’ and caregivers’ emotional responses to those stressors (e.g., feeling angry that one has diabetes or that one’s own child has diabetes, feeling “burned-out” due to the disease, feeling that one does not care enough about one’s diabetes or one’s child’s diabetes) [3,4,5]. Patient and parent-reported outcome in respect to DD is of great importance for well-being and, moreover, the level of DD may affect clinically important measures of glycaemic control.

Psychological distress may be influenced by additional factors, including demographics, coping style, family relationships, care demands and financial problems [6]. The coronavirus disease 2019 (COVID-19) pandemic caused by severe acute respiratory syndrome coronavirus 2 (SARS-CoV-2) is also a potential source of stressors and may cause psychological distress [7,8]. In Poland, first lockdown-type restrictions were introduced on 10–12 March 2020 and further strengthened on 25 March 2020: schools, universities and offices were closed, mass events were cancelled, non-family gatherings and traveling were strictly limited. Many employees and students at all education levels started to work from home, and access to long-term/chronic medical care became limited despite the switch to telemedical care. Primarily, we intended to longitudinally assess diabetes distress in patients who used different methods of glucose monitoring, but the pandemic emerged as a major disrupting factor and we used this opportunity to investigate whether and how the pandemic-related lockdown affected DD in children and adolescents with T1DM and their parents.

## 2. Materials and Methods

Patients with T1DM aged 8–18 years and their parents were eligible for this cohort study. We evaluated DD using age-specific Problems Areas in Diabetes questionnaires. We used PAID-Ch (Problem Areas in Diabetes—Child version) 11-item questionnaire for children 8–12 years, PAID-T (Problem Areas in Diabetes—Teen version) 14-item questionnaire for teens 12–18 years, P-PAID-Ch (Problem Areas in Diabetes—Parents of Children) 16-item questionnaire for children’ parents and P-PAID-T (Problem Areas in Diabetes—Parents of Teens) 15-item questionnaire for teens’ parents [9,10]. Problem areas covered by PAID questionnaires include emotional burden, diabetes therapy and regimen-specific burden, family and friends-related distress. Higher PAID score reflects a higher level of DD. The tool was validated in terms of psychometric properties in a number of populations, including European ones, and demonstrated good responsivity in intervention studies. For use in the following study, PAID questionnaires were translated from English into Polish and both versions were reverse checked for compatibility. Paper versions of PAID questionnaires were offered to participants during consultations between November 2019 and February 2020 (first PAID survey). Throughout the spring 2020 COVID-19 lockdown, the outpatient clinic activity was restricted to teleconsultations, so at the follow-up participants were asked to answer the same PAID questionnaires by phone-calls conducted by researchers (second PAID survey, April 2020). In addition, during the lockdown parents were asked to answer supplementary semi-open questions concerning the impact of the pandemic on diabetes care-related difficulties and worries (Supplementary Material, Questionnaires). The study was approved by the local bioethics committee (RNN/72/20/KE) and all included participants provided informed consent to participate in the surveys. 

### Statistical Analysis

Due to the differences in PAID questionnaires between children, teens and their respective parents (different number and wording of questions), each of those results was analysed separately. Continuous variables were summarized with median and interquartile ranges (median (Q1–Q3)), categorical variables with N (%)). Between-group comparisons were performed with Mann-Whitney’s *U* test while changes in each score were assessed with Wilcoxon’s test for dependent observations. Due to the small number of participants, detailed subgroup analyses with interactions of multiple factors were not performed. Associations between continuous variables were assessed with Spearman’s R correlations. For all tests, Alpha threshold for significance was set as 0.05.

## 3. Results

### 3.1. Study Group Characteristics

We included 76 patients (30 girls and 46 boys; 21 children and 55 teens; median age of participants 13.6 years (Q1–Q3: 11.8 to 15.2); T1DM duration 3.7 years (1.7 to 6.8)) and their parents. Recruitment flow chart is presented in Figure A1. Baseline patients’ characteristics is presented in Table 1. Majority of the group was treated with insulin pumps (continuous subcutaneous insulin infusion, CSII), the rest with multiple daily injections (MDI). At baseline, 42 patients (55.3%) used self-monitoring of blood glucose (SMBG), and others used continuous glucose monitoring (CGM).

### 3.2. Diabetes Distress before the Pandemic

Before the pandemic, median PAID score in children was 31 (26.0–42.0) and in their parents 52.0 (48.0–70.0). In teens it was 39.0 (30.0–47.0) and in teens’ parents 57.0 (46.0–65.0) (Table 1). PAID scores did not correlate with HbA1c but, for teens and their parents, were positively associated with BMI z-score (teens: R = 0.31, *p* = 0.0206; teen’s parents: R = 0.31, *p* = 0.022) (Table A1). Teenage girls presented significantly higher PAID score than boys (44.5 (40.0–50.5) vs. 34.0 (24.0–42.0), *p* = 0.003). Type of glucose monitoring (SMBG vs. CGM) or mode of insulin therapy (MDI vs. CSII) did not significantly affect PAID score neither in patients nor in parents (Table A2).

### 3.3. Diabetes Distress during Pandemic

In adolescents PAID score decreased significantly during the lockdown (median difference −3.0 [−11.0–3.0], *p* = 0.0183) (Figure 1, Table A3).

The PAID score decrease was evident in teenage girls (−7.0 (−17.0 to −2.5)) but not in boys (0 (−9.0–5.0), for comparison of change in girls vs in boys *p* = 0.028, Table 2). In children, PAID score did not change significantly (−3.0 (−14.0–7.0), *p* = 0.131). In parents of teens and in parents of children no significant change in PAID score was observed (teens’ parents: 3.0 (−9.0–10.0), *p* = 0.376; children’s parents: −5.0 (−9.0–1.0), *p* = 0.227).

The change in PAID score during lockdown was negatively correlated with the baseline PAID score in children (R = −0.57, *p* = 0.0065), teens (R = −0.55, *p* < 0.0001) and in teens’ parents (R = −0.32, *p* = 0.0166). There was a significant difference in PAID score change between children’s parents who did not report COVID−19-related worries (−10.5(−17-(−6))) compared with those who did report such worries (0 (−7(−8)), *p* = 0.021, Table 2).

Between the two assessments 22 patients (12 teens and 10 children) changed their glucose monitoring method from SMBG to CGM, starting to use FreeStyle Libre (FSL, Abbot Diabetes Care, Oxon, UK). In teens, these who switched to CGM did not present a directional difference in change in PAID score (N = 12; median difference 0 (−8.0–2.5)) compared to those who continued with only SMBG (N = 19; −7.0 (−17.0–1.0), *p* = 0.3397). In children who switched from SMBG to CGM the PAID score change was −4.0 (−10.0–0), but comparison with not-switchers was not possible, as only 1 child continued with SMBG only. The SMBG-to-CGM switch had no evident impact on difference between PAID score change in parents of teens (teen’s parents who switched to CGM: 6.5 (−8.5–9.0) vs parents who continued with SMBG: 5.0 (−6.0–14.0), *p* = 0.792); in parents of children who switched to CGM the PAID score change was −6.0 (−11.0–6.0); however, as with their children, the comparison with not-switchers was not possible.

## 4. Discussion

During the COVID-19 pandemic-related lockdown DD decreased in adolescents, especially in girls. Distress decline in teens may be possibly related to staying at home, where it might have been easier for them to obey diabetes management rules and avoid school-related distress. Our data show that this might be the case especially for those struggling the most with managing their diabetes and presenting high PAID score at baseline, as we found that in three of four studied subgroups (except parents of children <12 years) the higher the baseline level of DD, the more distress level decreased. In the studied group, this applied in particular to teenage girls as, before the COVID-19 pandemic, PAID score was higher in teenage females than in teenage males, and in teenage girls PAID score decrease during the pandemic was more evident than in male teens. In younger children, DD did not change significantly and this may be viewed as a consequence of substantial caregivers’ engagement in diabetes management in this age group, which to some degree might protect the younger group from being susceptible to pandemic-related changes in their DD. At the time of writing this article, we did not find studies directly comparing DD before and during pandemic. However, Passanisi et al. in their web-based survey reported that approach to the disease during pandemic was significantly less straightened in patients ≥12 years compared to the younger age group and that these older patients reported both more time spent on physical activities and fewer measurements of glucose levels than younger subjects [11]. Such behaviour might potentially contribute also to lower levels of DD reported by teenagers during the pandemic. Taking the above into account, one can suppose that young people with T1DM may require particular support (psychological and/or targeting their knowledge and skills related to diabetes management) when coming back to on-site school routine after the months of remote, on-line education.

We found that in the parents of young patients with T1DM who did not report COVID-19 related worries, DD decreased, which suggests that for a certain subgroup of caregivers the life conditions imposed by the pandemic could relieve the diabetes related burden. Parents seemed to be in this respect similar to adult T1DM patients, in whom it was observed that high DD was associated with higher level of worries about COVID-19 impact on their diabetes [8].

In the studied group, the use of CGM before the pandemic or introduction of CGM (FSL) between the surveys did not significantly impact the PAID score. However, as the median observation time in our study was 2 months and subgroups numbers were small, we are cautious regarding this observation. This deserves further study including more groups and longer observation time, as in larger studies including adults or youths it was shown that FSL reduced DD or increased self-reported treatment satisfaction [12,13].

The strengths of our study are that our data directly comparing DD before and during the lockdown in patients with T1DM (i) still seem to be unique, (ii) were based on validated measures (PAID questionnaires) and (iii) included a homogenous population with equal, reimbursed access to tertiary diabetes care. Significant limitations are that (i) the pre-pandemic recruited study groups were not numerous (and could not be increased) and (ii) the second survey was performed by phone. The latter was unavoidable, as during the lockdown the PAID survey could be done either remotely or not at all.

We know that we should “make no extrapolation” [14], but it is hard to dismiss that publications emerging until now show that, in many studied groups of persons with T1DM, the lifestyle changes imposed by the pandemic did not deteriorate glycaemic control, or even that certain glycaemic metrics improved [15,16,17,18,19,20]. Lack of glycaemic control deterioration seems to be in line with our observation that DD also did not aggravate.

## 5. Conclusions

In the studied community of young patients with T1DM and their parents we found that the COVID-19 pandemic-related lockdown did not aggravate diabetes distress. We suggest that the observed decrease in DD in T1DM teens, mainly in girls, during the lockdown should attract the attention of diabetes therapeutic teams and school-staff to take measures to prevent potential DD increase and to additionally support students with T1DM, in particular teenage females, when they come back to regular school on-site education.

## Figures and Tables

**Figure 1 ijerph-18-08527-f001:**
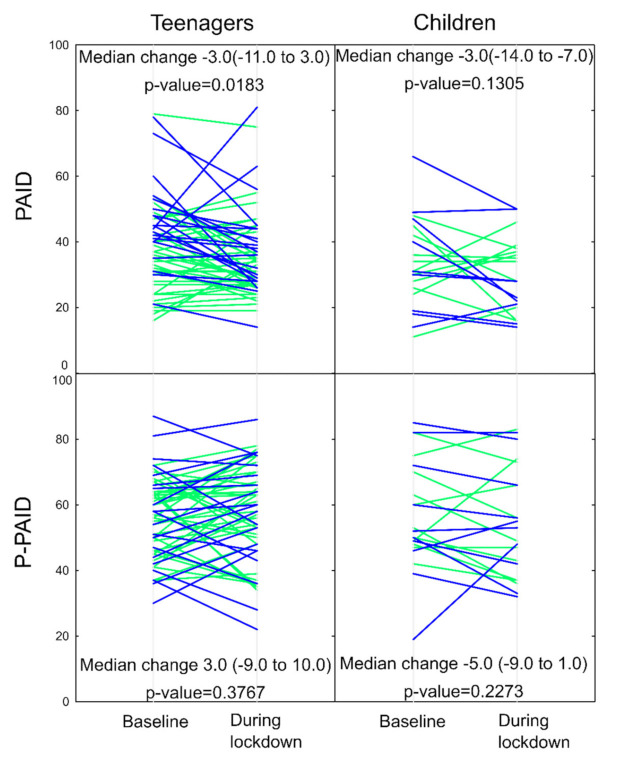
Diabetes-related distress expressed as PAID scores before (baseline) and during lockdown in children, teens and in their parents. Median change over time is shown for the whole group. Boys are denoted with light green, girls with blue. PAID—The Problems Areas in Diabetes in a Child or Teen; P-PAID—The Problems Areas in Diabetes in a Parent of Child or Teen.

**Table 1 ijerph-18-08527-t001:** Study group characteristics.

Characteristics		All Patients (N = 76)	Children (N = 21)	Teens (N = 55)	*p*-Value
Continuous Characteristics [Median (Q1–Q3)]					
Age(years)		13.6 (11.8–15.2)	10.1 (9.5–11.1)	14.4 (13.6–16.1)	N/C
T1DM duration (years)		3.7 (1.7–6.8)	3.0 (1.1–3.9)	4.5 (1.8–8.8)	0.0208
Time between 1st and 2nd PAID survey (days)		58.5 (45.5–98.5)	64.0 (45.0–93.0)	58.0 (46.0–99.0)	0.9028
HBA1c (%)		7.4 (7.2–8.1)	7.3 (6.8–7.5)	7.6 (7.2–8.7)	0.0786
HbA1c (mmol/mol)		57.4 (55.2–65)	56 (51–58)	60 (55–72)	0.0786
PAID		36.5 (29.0–46.0)	31.0 (26.0–42.0)	39.0 (30.0–47.0)	0.1383
P-PAID		56.0 (46.5–65.5)	52.0 (48.0–70.0)	57.0 (46.0–65.0)	0.9167
Nominal characteristics [N (%)]					
Gender structure (N (%) of boys))		46 (60.5%)	11 (52.4%)	35 (63.6%)	0.4356
T1DM duration ≤6 months		8 (10.5%)	4 (19%)	4 (7.3%)	0.2058
Body weight category (based on BMI z-score)	Normal	60 (80%)	19 (90.5%)	41 (76.9%)	0.3667
	Overweight	8 (10.7%)	1 (4.8%)	7 (13%)	
	Obesity	7 (9.3%)	1 (4.8%)	6 (11.1%)	
Type of insulin therapy	MDI	18 (23.7%)	15 (27.3%)	3 (14.3%)	0.3663
	CSII	58 (76.3%)	40 (72.7%)	18 (85.7%)	
Type of glucose monitoring	SMBG	42 (55.3%)	11 (52.4%)	31 (56.4%)	0.8005
	CGM	34 (44.7%)	10 (47.6%)	23 (43.6%)	
Responding parent	Mother	64 (84.2%)	19 (90.5%)	45 (81.8%)	N/C
	Father	5 (6.6%)	1 (4.8%)	4 (7.3%)	
	Both parents	1 (1.3%)	0 (0%)	1 (1.8%)	
	Missing data	6 (7.9%)	1 (4.8%)	5 (9.9%)	

T1DM, type 1 diabetes mellitus; PAID-Ch/PAID-T, The Problems Areas in Diabetes in a Child or Teen; P-PAID-Ch/T, The Problems Areas in Diabetes in a Parent of Child or Teen; T1DM, type 1 diabetes mellitus; HbA1c, glycated hemoglobin A1c; BMI z-score, body mass index standard deviation score; MDI, multiple daily injections; CSII, continuous subcutaneous insulin infusion i.e., pump therapy (pumps used by participants were MiniMed Veo or 640G, Medtronic MiniMed Inc.; AccuChek Performa Combo, Roche Diabetes Care); SMBG, self-monitoring of blood glucose; CGM, continuous glucose monitoring (systems used by participants were FreeStyle Libre, Abbot Diabetes Care; Enlite sensors integrated with Veo or MiniMed 640G pump; Dexcom G5/G6, Dexcom); N/C—not calculated.

**Table 2 ijerph-18-08527-t002:** Patients’ and parents’ change of PAID score (2nd PAID, i.e., during pandemic vs, 1st PAID, i.e., at baseline) in relation to clinical characteristics presented as medians (Q1–Q3) for subgroup comparisons and as R Spearman rank correlations for continuous variables. Pandemic-related difficulties and worries were assessed based on a semi-open questionnaire concerning the impact of the pandemic on diabetes care (attached in Appendix A). *p*-values for statistically-significant comparisons were bolded.

Clinical Characteristics		PAID-Ch Change	*p*-Value	P-PAID-Ch Change	*p*-Value	PAID-T Change	*p*-Value	P-PAID-T Change	*p*-Value
Subgroup Comparisons. Median (Q1−Q3)									
Sex	Male	0 (−10 to 9)	0.121	−7 (−14 to 6)	0.397	0 (−9 to 5)	**0.028**	3 (−6 to 10)	0.930
	Female	−4 (−17 to −2)		−4.5 (−7 to 1)		−7 (−17 to −2.5)		2.5 (−11.5 to 10.5)	
T1DM duration	>6 months	−3 (−16 to 1)	0.394	−4 (−7 to 6)	**0.022**	−4 (−13 to 2)	0.277	2 (−9 to 10)	0.065
	<6 months	2 (−7 to 8.5)		−12.5 (−15.5 to −9)		0.5 (−4.5 to 11)		12 (5 to 20)	
Glucose monitoring method	SMBG	−4 (−14 to 0)	0.217	−7 (−11 to 6)	0.724	−6 (−13 to 2)	0.592	5 (−6 to 10)	0.541
	CGM	2.5 (−16 to 9)		−4.5 (−7 to 0)		−2.5 (−11 to 3.5)		1.5 (−9 to 10)	
Type of insulin therapy	MDI	−10 (−17 to 9)	0.801	−14 (−17 to 1)	0.246	−4 (−7 to 5)	0.570	2 (−14 to 10)	0.533
	CSII	−2.5 (−14 to 7)		−5 (−7 to 6)		−2 (−13 to 2)		3 (−5.5 to 10.5)	
COVID-19 pandemic-related difficulties	Yes	−3 (−10 to 0)	0.859	−4 (−7 to 6)	0.354	−6.5 (−13 to 4)	0.410	2 (−12 to 7.5)	0.411
	No	−2.5 (−15.5 to 7.5)		−6.5 (−11.5 to 0.5)		−1 (−11 to 2)		5 (−6 to 10)	
COVID-19 pandemic-related worries	Yes	−3 (−16 to 7)	0.938	0 (−7 to 8)	**0.021**	−4.5 (−11 to 1)	0.298	2.5 (−6 to 10)	0.921
	No	−2.5 (−10 to 7)		−10.5 (−17 to −6)		−1 (−8 to 8)		7 (−11 to 10)	
Association with continuous variables. Spearman R correlations									
Age (years)		R = −0.12	0.594	R = −0.47	**0.032**	R = −0.03	0.851	R = 0.02	0.908
T1DM duration (years)		R = −0.08	0.722	R = 0.11	0.645	R = 0.19	0.160	R = 0.01	0.942
1st PAID		R = −0.57	**0.007**	R = 0.41	0.064	R = −0.55	**0.0001**	R = −0.06	0.662
1st P-PAID		R = −0.36	0.111	R = −0.19	0.402	R = −0.02	0.895	R = −0.32	**0.017**
BMI z-score		R = −0.19	0.408	R = −0.13	0.56	R = 0.03	0.832	R = −0.01	0.927
HbA1 (%)		R = −0.39	0.109	R = −0.46	0.052	R = 0.09	0.518	R = 0.23	0.102
P-PAID change		R = −0.05	0.821	NA	NA	R = 0.13	0.353	NA	NA

## Data Availability

The datasets generated and analysed during the present study are available from the corresponding author on request.

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
