# Peer review of "Diabetes Related Distress in Children with Type 1 Diabetes before and during the COVID-19 Lockdown in Spring 2020"

_ijerph, 2021, doi:10.3390/ijerph18168527_

Round 1

Reviewer 1 Report

The topic is current, timely and interesting. However, sample size and experimental design is very weak for any interpretation. Authors stated:

We conclude that in the studied community of young patients with T1DM and their par- 208 ents the COVID-19 pandemic-related lockdown did not aggravate diabetes distress.

This is very hard to conclude with the attached manuscript, sample size and experimental design. They need much higher sample size numbers, more direct measures rather than a simple survey. They need to expand their study possibly other areas to be able to generalize their findings as well. 

Author Response

To Reviewer 1

Dear Reviewer thank you very much for dedicating your time for the revision of our manuscript, for your interest in the subject and for pointing the timeliness of the topic. We appreciate your valuable suggestions. We address the issues below your revision text.

The topic is current, timely and interesting. However, sample size and experimental design is very weak for any interpretation. Authors stated:

We conclude that in the studied community of young patients with T1DM and their par- 208 ents the COVID-19 pandemic-related lockdown did not aggravate diabetes distress.

This is very hard to conclude with the attached manuscript, sample size and experimental design. They need much higher sample size numbers, more direct measures rather than a simple survey. They need to expand their study possibly other areas to be able to generalize their findings as well. 

We agree that the sample size is not big; it was a group that we planned to extend for further assessment of DD level and DD change over time, however the pandemic emerged as an unpredictable event and forced us to close the participants recruitment. As we state in the manuscript (lines 59-63)  “Primarily, we intended to longitudinally assess diabetes distress in patients who used different methods of glucose monitoring, however the pandemic emerged as a major disrupting factor and we used this opportunity to investigate whether and how the pandemic-related lockdown affected DD in children and adolescents with T1DM and their parents.”  The assessment of this group just several weeks before the COVID-19 pandemic gave us a unique possibility to compare the level of diabetes distress (DD) over the period when it was unlikely to have been influenced by other factors than the pandemic and lockdown themselves. It was made possible especially as we did not find a significant influence of the method of glucose monitoring or of the change of glucose monitoring method on DD (we compared two methods of glucose monitoring used by the participants: (i) monitoring based only on blood glucose meter measurements, i.e. self-monitoring of blood glucose [SMBG ] vs (ii) the use of modern continuous glucose monitoring [CGM], namely flash glucose monitoring [FGM]). It is not in any way possible for us to increase the sample size recruited before the pandemic as we included all the patients and parents who filled in PAID questionnaires before the Covid-19 related lockdown. Even though the study group is not big we applied statistical methods that were proper for such a sample size and we believe that our results are reliable and may be valuable for clinicians. Moreover, the initial questionnaires were collected in an unbiased way in our centre during routine visits. Therefore, the risk of bias (other than non-response bias) affecting the generalizability of our results to the centre`s population is small.

Regarding the measures of diabetes distress, we used PAID questionnaires addressed for particular patient age-groups and for their parents. PAID questionnaires are validated tools for diabetes distress assessment for these populations. For our study design these relatively simple questionnaires covering a broad spectrum of diabetes distress aspects were very suitable, as they were easy for filling in paper versions before the pandemic as well as with phone calls during the lockdown.

We agree that, as the sample size was small, we can not generalize and this is why we are cautious and formulate our conclusion not as statement/s but only as a suggestion. We have tempered the Conclusion paragraph accordingly.

Reviewer 2 Report

In the paper named ”Diabetes related distress in children with type 1 diabetes before 2 and during the COVID-19 lockdown in spring 2020” author make a of diabetes-related distress (DD) in young patients with type 1 diabetes mellitus (T1DM) 17 and in their parents before and during the national COVID-19-related lockdown when schools operated on-line. Their results shown that

COVID-19 pandemic-related lockdown was associated with decrease in DD in 26 teens with T1DM while no significant change in DD was observed in children or parents.  The author say that DD decrease in teens 27 during the pandemic should attract attention because it can be a potential “rebound” of DD related to return to regular on- site school routine.

Due the relevance and quality of the work, there isonly one question and one comment.

Does the author do not find differences between girls and boys? This question is in the sense that girls should be more careful and concerned.

Author Response

To Reviewer 2

Dear Reviewer thank you very much for dedicating your time for the revision of our manuscript and  for pointing its relevance. We appreciate your valuable suggestion which we address below your revision text.

In the paper named ”Diabetes related distress in children with type 1 diabetes before 2 and during the COVID-19 lockdown in spring 2020” author make a of diabetes-related distress (DD) in young patients with type 1 diabetes mellitus (T1DM) 17 and in their parents before and during the national COVID-19-related lockdown when schools operated on-line. Their results shown that

COVID-19 pandemic-related lockdown was associated with decrease in DD in 26 teens with T1DM while no significant change in DD was observed in children or parents.  The author say that DD decrease in teens 27 during the pandemic should attract attention because it can be a potential “rebound” of DD related to return to regular on- site school routine.

Due the relevance and quality of the work, there is only one question and one comment.

Does the author do not find differences between girls and boys? This question is in the sense that girls should be more careful and concerned.

As you noticed, in the studied group we found differences between teenage girls and boys:

- lines 120-121: Before the pandemic “[…] Teenage girls presented significantly higher PAID score than boys (44.5 [40.0-50.5] vs 34.0 [24.0-42.0], p=0.003).”

- …lines 136-137: “The PAID score decrease was evident in teenage girls (-7.0 [-17.0 – (-2.5)]) but not in boys (0 [-9.0-5.0], for comparison of change in girls vs in boys p = 0.028, Table 2).” Therefore, it seems that girls were indeed more concerned and stressed about their diabetes than boys. However, their diabetes distress also dropped more than boys` during the lockdown period (we added colours in Figure 1 to differentiate between females and males). We can only speculate that in case of girls, school and peer-related pressure might have contributed more to diabetes distress.

According to what you suggest, now we underlie this difference between male and female teenagers also in the Conclusions section. 

Round 2

Reviewer 1 Report

in normal circumstances, I would not accept this revision, however it is understandable that  pandemic make everything hard and limiting higher quality study designs etc.

Author Response

Thank you for your suggestion.

This manuscript is a resubmission of an earlier submission. The following is a list of the peer review reports and author responses from that submission.